# Genetic and Clinical Spectrum of GNE Myopathy in Russia

**DOI:** 10.3390/genes13111991

**Published:** 2022-10-31

**Authors:** Aysylu Murtazina, Sergey Nikitin, Galina Rudenskaya, Inna Sharkova, Artem Borovikov, Peter Sparber, Olga Shchagina, Alena Chukhrova, Oksana Ryzhkova, Olga Shatokhina, Anna Orlova, Vasilisa Udalova, Ilya Kanivets, Sergey Korostelev, Alexander Polyakov, Elena Dadali, Sergey Kutsev

**Affiliations:** 1Research Centre for Medical Genetics, Moscow 115478, Russia; 2Laboratory Genomed LTD, Moscow 107014, Russia; 3Federal State Budgetary Educational Institution of Further Professional Education “Russian Medical Academy of Continuous Professional Education” of the Ministry of Healthcare of the Russian Federation, Moscow 125993, Russia

**Keywords:** GNE myopathy, Nonaka myopathy, hereditary inclusion body myopathy, atypical cases

## Abstract

GNE myopathy (GNEM) is a rare hereditary disease, but at the same time, it is the most common distal myopathy in several countries due to a founder effect of some pathogenic variants in the *GNE* gene. We collected the largest cohort of patients with GNEM from Russia and analyzed their mutational spectrum and clinical data. In our cohort, 10 novel variants were found, including 2 frameshift variants and 2 large deletions. One novel missense variant c.169_170delGCinsTT (p.(Ala57Phe)) was detected in 4 families in a homozygous state and in 3 unrelated patients in a compound heterozygous state. It was the second most frequent variant in our cohort. All families with this novel frequent variant were non-consanguineous and originated from the 3 neighboring areas in the European part of Russia. The clinical picture of the patients carrying this novel variant was typical, but the severity of clinical manifestation differed significantly. In our study, we reported two atypical cases expanding the phenotypic spectrum of GNEM. One female patient had severe quadriceps atrophy, hand joint contractures, keloid scars, and non-classical pattern on leg muscle magnetic resonance imaging, which was more similar to atypical collagenopathy rather than GNEM. Another patient initially had been observed with spinal muscular atrophy due to asymmetric atrophy of hand muscles and results of electromyography. The peculiar pattern of muscle involvement on magnetic resonance imaging consisted of pronounced changes in the posterior thigh muscle group with relatively spared muscles of the lower legs, apart from the soleus muscles. Different variants in the *GNE* gene were found in both atypical cases. Thus, our data expand the mutational and clinical spectrum of GNEM.

## 1. Introduction

GNE myopathy (GNEM) was first described independently by Nonaka in 1981 as distal myopathy with rimmed vacuoles [1] and by Argov and Yarom in 1984 as rimmed vacuole myopathy sparing the quadriceps [2].

GNEM is a distal young adult-onset myopathy with autosomal recessive inheritance caused by biallelic loss-of-function variants in the *GNE* gene. That gene encodes UDP-N-acetylglucosamine 2-epimerase/N-acetylmannosamine kinase, a rate-limiting enzyme for biosynthesis of N-acetylneuraminic acid, which is a major type of sialic acid in humans [3]. Hyposialylation leads to changing in many biologic processes, especially in cell adhesion and signal transduction in various organs. It was showed that null mice are embryo lethal, and no patient with two null variants was reported [4]. Thus, causative variants in the *GNE* gene must preserve some residual enzyme activity.

To date, several large cohorts of patients with GNEM were described [5,6,7,8,9]. More than 250 pathogenic and likely pathogenic variants in the *GNE* gene are known according to the Human Gene Mutation Database [10]. For several of them, a founder effect was shown in different populations: Israelis, Romans, Japanese, Caucasians, Indians, and Chinese [6,7,8,11,12,13].

GNEM has a distinct phenotype characterized by spared vastus lateralis muscle even in the advanced stages of the disease, when all the other leg muscles are completely replaced by fat [14]. Due to this selective muscle involvement, patients with GNEM have a quite recognizable gait when they “throw” the lower leg forward when walking. Only a small number of cases with unusual features such as significant paravertebral muscle atrophy [15], atypically more prominent involvement of calf muscles [16], or asymmetric hand weakness [17] were reported. The severity of myopathy varies significantly in all cohorts of patients, even in affected members of the same family [16]. It was shown that the severity of the disease may partially be attributable to the specific *GNE* genotype [18]. Other factors that may contribute to the severity of the disease remain unknown.

Recently, several works about extra-muscular manifestations of GNEM were published [19,20]. According to them, idiopathic thrombocytopenia and sleep apnea syndrome are observed in GNEM patients more frequently than in the general population.

In the current paper, we describe the largest cohort of patients with GNEM from Russia. Two cases in our cohort expand the phenotypic variability of GNEM.

## 2. Subjects and Methods

### 2.1. Patient Cohort

Our study is a retrospective cohort study of patients with genetically confirmed GNEM. From 2017 to 2021, 31 patients from 27 unrelated non-consanguineous families were diagnosed with GNEM by genetic testing. Nine patients out of them were described previously [21]. The cohort of patients included 9 males (29%) and 22 females (71%). Median age of GNEM diagnosis was 32 (29; 37) years. Three of the unrelated patients were from the Bashkir ethnic group, and the rest of the patients were of Russian origin. Twenty-two patients filled out the evaluation form of the GNE Myopathy Functional Activity Scale (GNEM-FAS) [22]. Neurological examination was performed in 25 patients from 21 unrelated families. An in-person clinical examination was impossible in 6 patients, and we could not collect enough clinical data about them. In 20 patients, we carried out the manual muscle testing of 30 muscles by using the Medical Research Council’s score (30 MMT). Laboratory analyses included platelet count and serum biochemistry.

Needle electromyography (EMG) (Dantec Keypoint G2, Denmark and Neuro-MEP-micro, Neurosoft, Ivanovo, Russia) was recorded in 19 patients. Different muscles were tested, but preferable muscles were deltoideus, extensor digitorum, vastus lateralis, tibialis anterior, and gastrocnemius medialis.

Magnetic resonance imaging (MRI) of leg muscles (axial native T1-weighted and axial T2-STIR images) were obtained from 10 patients (1.5T magnetic resonance scanner, Siemens, Munich, Germany). Fat infiltration of skeletal muscles was assessed on the T1-weighted images using the scale published by Mercuri et al. [23], where stage 0 refers to a normal appearance, stage 1 reflects mild moth-eaten appearance in small areas, stage 2 describes the changes in a larger volume of muscle tissue like increased signal intensity in <60% of the muscle, stage 3 represents washed-out appearance and increased signal intensity in >60% of muscle volume, and stage 4 refers to an entire muscle tissue involvement.

### 2.2. Genetic Analysis

The DNA analysis for the patients was carried out using paired-end sequencing (2 × 75 bp) on an IlluminaNextSeq 500 sequencer. The libraries were constructed using the IlluminaTruSeq^®^ ExomeKit (San Diego, CA, USA). The detected variants were annotated according to the HGVS nomenclature: http://varnomen.hgvs.org/recommendations/DNA, accessed on 31 August 2022, version 2.15.11.

The sequencing data were analyzed using a standard Illumina pipeline https://basespace.illumina.com, accessed on 31 August 2022. Mean coverage was ×102.4, with 3% of fragments with less than ×10 coverage.

The following predictive algorithms to analyze the pathogenicity of the variants were used: SIFT (Sort Intolerant From Tolerant Human Protein) [24], UMD-Predictor [25], and PolyPhen-2 (Polymorphism Phenotyping v2) [26].

Automatic Sanger sequencing was carried out using ABIPrism 3100xl Genetic Analyzer (Applied Biosystems, Foster City, CA, USA) according to the manufacturer’s protocol. Primer sequences were chosen according to the NM_001128227.3.

The detected variants were classified as pathogenic, likely pathogenic or variants of uncertain significance according to the ACMG/AMP guidelines [27].

## 3. Results

### 3.1. Mutation Analysis

GNEM was confirmed by next-generation sequencing (NGS) studies and Sanger sequencing in 31 patients from 27 unrelated families from different districts of the Russian Federation. A total of 22 different variants were found. Twelve previously reported variants detected in our cohort were missense variants. Some of them (p.Asp207Val, p.Val603Leu, p.Ile618Thr, p.Ala662Val) are frequent variants in several populations [6,8,12,13].

Ten out of 22 variants were novel, which accounted for 39% of all alleles (Figure 1a). Among novel variants, six missense variants (p.Glu33Asp, p.Ala57Phe, p.Tyr187Cys, p.Cys203Ser, p.Gly607Arg, p.Ala669Ser), two frameshift variants (p.Met263CysfsTer4, p.Ser614ArgfsTer6) (Figure 1b), and 2 gross deletions were found (Table 1). All novel variants except for the p.Ala57Phe variant, were identified in a compound heterozygous state with other variants. According to the ACMG guidelines, two novel frameshift variants were classified as pathogenic, and two missense variants, c.99G>C (p.(Glu33Asp)) and c.169_170delGCinsTT (p.(Ala57Phe)), as likely pathogenic. The other novel variants were classified as variants of uncertain significance (Table 1). Two gross deletions were suspected by the whole exome sequencing with approximate breakpoints at chr9:36245968-36265549del and chr9:36227200-36234171del (hg19), leading to complete deletion of exons 1–3 and 5–7, respectively.

The variant c.169_170delGCinsTT (p.(Ala57Phe)) was found in a homozygous state in four families (six patients) and in a compound heterozygous state with other variants in three unrelated cases. All these patients were originally from the three neighboring areas of European Russia (Bashkortostan Republic, Mari El Republic, Tatarstan Republic) (Figure 1c). Three of those patients were Bashkirs, and the rest of them were Russian. It should also be noted that the novel variant c.169_170delGCinsTT (p.(Ala57Phe)) is the second common variant in our cohort (20% of all alleles) (Figure 1a). This variant is absent in the Genome Aggregation Database as well as in the database of genetic variants found in residents of the Russian Federation [28,29].

In total, 18 missense variants including 6 novel variants were identified in our cohort. Half of the variants were located in the epimerase or kinase coding regions.

Two previously reported pathogenic variants c.1853T>C (p.(Ile618Thr)) and c.1760T>C (p.(Leu587Ser)) were found in a homozygous state in our cohort. Three unrelated patients were homozygous for the c.1853T>C (p.(Ile618Thr)) variant, which was reported many times and has a founder effect in Roma and Rajasthan patients [8,13]. These three patients identified themselves as Russian, and two of them were from nearby districts of the Russian Federation (Omsk and Kurgan regions) (Figure 1c). Moreover, the c.1853T>C (p.(Ile618Thr)) variant was detected in a heterozygous state in 6 unrelated patients, and thus, that is the most common allele in Russian GNEM patients (22% of all alleles) (Figure 1a).

Three unrelated patients were homozygous for the c.1760T>C (p.(Leu587Ser)) that was reported once in a patient of Caucasian ancestry [30]. Two of these patients were from distinct parts of the Russian Federation (St Petersburg and Sverdlovsk region). The third patient was born in another country, Kazakhstan, but identified herself as Russian (Figure 1c).

### 3.2. Clinical Characteristics

Clinical examination was performed in 25 patients from 21 unrelated families (Table 2). In our cohort, the number of isolated cases was 18 and there were 3 familial cases with 2 or 3 affected siblings (families 4, 12, 19). All patients except for two had a similar clinical picture. There was a rather narrow range of the age at time of evaluation and the age of disease onset. The median age at examination was 32 (29; 37) years, median age of onset was 24 (20; 25) years. The first symptoms were bilateral foot drop in 16 patients (64%), pain in leg muscles or low back pain in 4 patients (16%). Three patients reported difficulties in climbing stairs as the first symptom. Upper limb muscle weakness, one wasting of leg muscles or inability to stand on the toes were observed once as the first symptoms in different patients.

Seven patients (28%) were non-ambulant at the time of examination, and the median age of ambulation loss was 32 (5; 28; 34) years. All ambulant patients had gait disturbances, and most of them (89%) had GNEM specific gait pattern. Neurological examination showed that the foot and the lower leg muscles were atrophied uniformly. Nineteen patients (76%) had hand and/or forearm muscle atrophy (Figure 2). Thirty MMT score and GNEM-FAS were measured in 20 (80%) and 22 (88%) patients, respectively. The range of scores in our cohort was broad from 14 to 138 in 30 MMT score and from 2 to 97 in GNEM-FAS. In all patients, foot and lower leg muscles were most severely affected. In one quarters of patients, proximal limb muscles were also affected, except for quadriceps muscles that were wasted only in two patients. Neck muscle weakness was observed in 9 (36%) patients. Ankle reflexes were reduced uniformly, and patellar reflexes were spared in 7 (29%) patients. Two patients had an extremely low score of GNEM-FAS of less than 10 out of 100 points. One of these patients had dysphagia. Two other patients from our cohort had atypical clinical features and they are reported in more detail below.

Laboratory data apart from creatine kinase (CK) level were usually normal. CK level was measured for 22 patients; in 16 out of them (73%), it was elevated from 250 to 1200 U/L (Table 2). The number of platelets in the blood was known for 13 patients, and in 12 cases it was normal. One patient (No. 21.1) had thrombocytopenia with a platelet count of 126 × 10^9^/L (nl 180–450 × 10^9^/L).

Needle EMG was performed for 19 patients. Four patients (No 10.1, 17.1, 18.1, and 20.1) were initially diagnosed with spinal muscular atrophy due to neurogenic changes on needle EMG. In all these cases, vastus lateralis was studied, and the amplitude and the duration of motor unit potentials were increased in that muscle. For 2 patients, needle EMG was repeated, and in most clinically affected muscles, myogenic motor unit potentials were found. In the other 15 patients, myogenic pattern was revealed primarily. In 12 cases (63%), moderate or severe abnormal spontaneous activity was found, mostly in the form of positive sharp waves and fibrillation potentials.

Eight patients underwent muscle MRI, and in six of them a typical pattern of muscle involvement was observed. Vastus lateralis is spared almost in all patients (6 cases); in one case (patient No. 7.1), it was mildly involved due to long disease duration (19 years) and in patient No. 8.1, fat replacement of vastus lateralis was pronounced as an atypical presentation of GNEM. One patient with typical clinical features of GNEM had short disease duration (patient No. 21.1) and few muscles were replaced by fat. More affected muscles were lower leg muscles excluding tibialis posterior, and adductor magnus, semimembranosus, and sartorius on thigh level (Figure 3).

### 3.3. Atypical Cases

We observed two cases which had atypical clinical and visualization data (patients No. 8.1 and 18.1). The first case was a 23-year-old female patient (Figure 4) with prominent atrophy of all limb muscles, including quadriceps femoris bilaterally (Figure 4c). On muscle MRI, vastus lateralis and rectus femoris demonstrated a pattern of multiple striped signal abnormalities (like a “sandwich” sign) that is specific for COL6-related myopathies (Figure 4e). Additionally, the patient had contractures of hand joints and keloids (Figure 4b,d). She underwent whole exome sequencing with referral diagnosis of atypical COL6-related myopathy. However, it revealed the previously reported heterozygous variant c.1853T>C (p.(Ile618Thr)) in the *GNE* gene and suspected deletion of exons 5–7 of the *GNE* gene predicted by reads count.

The patient was a single affected family member (Figure 4a). Her first symptoms at the age of 20 years were bilateral foot drop appearing by frequent stumbling followed by steppage gait. In 2 years, she had marked progression of the disease and since the age of 22 years she needed support when walking. Neurological examination at the age of 23 years showed atrophy of the muscles of distal and proximal upper and lower extremities, weakness of leg muscles (2/5), arm muscles (3/5), neck extensor muscles (3/5), contractures of the left-hand interphalangeal joints, and bilateral ankle tendon contraction. Her CK level was 103 U/L (nl 25–200 U/L). The patient had unexplained persistent proteinuria 0.5 g/L (nl < 0.033 g/L). The number of platelets in the blood was normal. Ultrasound of the abdominal cavity and retroperitoneum revealed no changes, as well as an ECG. EMG showed myopathic changes with severe spontaneous activity presented in the form of fibrillation potentials, positive waves, and complex repetitive discharges.

The second patient with atypical GNEM was a 34-year-old man (Figure 5) who was initially diagnosed with spinal muscular atrophy because of a very unusual clinical picture and non-relevant EMG results. At the time of examination, the patient had complaints of discomfort, heaviness, and pain in the legs, especially after prolonged physical exercises, weakness of the left arm muscles, change in his gait with feeling of incomplete support during a walk. From the anamnesis, it is known that about 6 years ago, he began to worry about pain in his legs, and then he noticed that it became hard to run. After 4 years, he noticed a loss of left hand and left leg muscles. Neurological examination revealed hypotrophy of the left leg muscles (Figure 5b), moderate asymmetrical hypotrophy of foot and hand muscles, more prominent on the left side, selective muscle weakness (iliolumbar muscles 3/5, gluteal muscles 4/5 bilaterally, left lower leg muscles 4/5, right foot muscles 2/5, left foot muscles 1/5). Deep arm tendon reflexes were reduced, as well as the left patellar reflex. The patient demonstrated signs of waddling gait without steppage, and he could stand on his toes and on his heels.

The CK level was increased up to 950 U/L (nl 25–200 U/L). The number of platelets, as well as other laboratory tests, were normal. EMG was performed three times. The first two procedures concluded that the patient had neurogenic changes in deltoideus and vastus lateralis. However, the third procedure of EMG studied the most affected muscles and revealed myogenic changes with moderate spontaneous activity.

The muscle MRI was performed at the age of 34 years (Figure 5c). Symmetrical severe changes were revealed in the muscles of the posterior compartment on the thigh level. However, in the lower legs, only the soleus muscle was predominantly affected, tibialis anterior, and gastrocnemius medialis were affected asymmetrically and much less.

Before the NGS study, the *SMN1* gene copy analysis was performed, it showed a normal count of the gene copies. Then, the patient was referred to the NGS study, and it revealed two earlier reported heterozygous missense variants in the *GNE* gene c.829C>T (p.(Arg277Trp)) and c.1618C>T (p.(His540Tyr)) [12,31]. Sanger sequencing showed that the patient’s parents were heterozygous carriers of each variant (Figure 5a).

## 4. Discussion

Numerous large cohorts of patients with GNEM were described around the world [6,7,8,9,12,13]. In our study, we described a group of patients from Russia who were diagnosed with GNEM in the period from 2017 to 2021. We analyzed the mutational spectrum in 27 patients. Just as in our study, in all the previously described cohorts, most of the variants were missense, and cases with biallelic null variants have never been reported yet [12,13,32]. All missense variants detected in our patients are distributed equally in the epimerase or kinase domain coding regions.

In our study, ten new variants were identified, of which c.169_170delGCinsTT (p.(Ala57Phe)) is often found in several nearby regions of Russia, and at the same time, this variant is not represented in existing population databases [28,29]. We plan to further study the population characteristics of this variant in cohorts of healthy people from these regions. This variant represents a deletion and an insertion of two nucleotides, which leads to the amino acid substitution of alanine to phenylalanine at position 57. Previously, another missense variant was reported at this amino acid position in one patient with typical clinical features of GNEM [33]. The variant c.169_170delGCinsTT (p.(Ala57Phe)) is the second most frequent variant in the cohort of patients from Russia, and it is found in a homozygous state in four families.

The most frequent variant in our cohort was c.1853T>C (p.(Ile618Thr)), and that was reported earlier many times in the Roma and Rajasthan populations [8,13]. However, all our patients who have this variant in a homozygous or heterozygous state have Russian ancestry and come from different parts of Russia.

Of the new variants identified in our patients, two are large deletions of several exons of the *GNE* gene. These deletions were detected during in the NGS study, and we were not able to validate them by other molecular genetic methods, which was a limitation of our study. Nevertheless, large rearrangements in the *GNE* gene have been reported several times [34,35], and the possible occurrence of CNVs should be considered in cases of myopathy with a heterozygous variant in the *GNE* gene.

Our cohort of patients was homogeneous in age at the time of examination, but the severity of clinical features presented by patients strongly varied, as can be seen from the range of GNEM-FAS and 30-MMT scores. Genotype–phenotype correlations were only partially detected [18]. It is known that patients carrying at least one allele with the c.620A>T (p.(Asp207Val)) variant have a later onset and a milder phenotype [12,32]. At the same time, there is a rather pronounced intra- and inter-familial phenotype heterogeneity even in cohorts of patients with the identical homozygous variant [8,36]. Unfortunately, our data were insufficient for genotype–phenotype analysis due to limited patient number.

In five patients with a new frequent variant c.169_170delGCinsTT (p.(Ala57Phe)) and known clinical status, a large range of clinical severity was noted. The age of onset corresponds to the age reported in patients with GNEM previously [37]. However, even siblings from the same family (No. 12) with a homozygous variant have a significant difference on the scores of GNEM-FAS, which could not be explained by the age of affected family members. Among patients who were homozygous for this variant, one patient (No. 13.1) had an extremely severe course of the disease with a non-ambulatory condition from the age of 30, dysphagia, and respiratory dysfunction. The severity of her disease corresponded to the development of respiratory dysfunction, as previously it was shown in GNEM patients [38].

There were no patients with a late onset of the disease after 30 years in our cohort, the latest age of onset was 29 years, and the youngest was 13 years. In the literature, cases with the age of onset at 10–12 years can be found [9,32]. At the time of examination, 7 out of 25 patients were non-ambulatory, and they were mostly female, which corresponds to observations of a more severe course of the disease in women [37]. The earliest wheelchair-bound age was 26 years, and the oldest one was 37 years.

Three quarters of our patients had significant atrophy of the hand and foot muscles, which is uncommon in distal myopathies and usually allows to distinguish distal myopathies from patients with neurogenic disorders. Atrophy of hand muscles was reported in most GNEM cases [37], but we focus on this sign, since in our opinion, it often confuses doctors and leads to a wrong diagnosis of hereditary neuropathy or distal spinal muscular atrophy. Moreover, sometimes needle EMG of clinically intact vastus lateralis may not show a myogenic pattern, but on the contrary, show an increase in the parameters of motor unit potentials due to muscle hypertrophy, which leads to an erroneous interpretation of these changes as neurogenic. Additionally, cases with neurogenic changes in other muscles were reported earlier [39,40]. However, it has recently been suggested that motor axonal neuropathy happens in GNEM [40]. Four patients from our cohort were initially wrongly diagnosed with distal spinal muscular atrophy or hereditary neuropathy. Our data indicate predominant diagnostic value of clinical evaluation and the importance of choosing proper muscles for needle EMG.

In a recent article, extra-muscular manifestations of GNEM including idiopathic thrombocytopenia were reported [19]. In our cohort, the platelet count was measured in 13 patients, and in 92% of cases, it was normal; only one patient had thrombocytopenia. We have no information about sleep apnea syndrome (another GNEM extra-muscular manifestation) in our patients.

In all patients who underwent muscle MRI, apart from two atypical cases, the pattern of leg muscle involvement was classic with uniformly affected lower leg muscles and almost all muscles of thigh posterior compartment in advanced cases [41]. Vasti muscles were severely affected in only one atypical case (patient No. 8.1), and a pattern-like “sandwich” sign was noted. Contractures of interphalangeal joints and keloid scars resembled an atypical collagenopathy, type 6 [42,43]. However, no variants in the COL6 genes were detected, but a deletion of 5–7 exons in the *GNE* gene and the previously reported missense variant c.1853T>C (p.(Ile618Thr)) were found. Undoubtedly, additional genetic studies are needed in this case to explain such a peculiar clinical picture. However, the patient was no longer available due to long-distance residence. Nevertheless, the patient’s gait and the combination of the myogenic pattern on EMG with distal atrophy allow us to consider the detected variants in the *GNE* gene as the cause of the disease. We have not met similar atypical clinical cases in the literature.

In the second atypical case (patient No. 18.1), the peculiarity of muscle MRI was shown with pronounced changes in the posterior thigh muscle group, while the muscles of the lower legs, apart from soleus, were relatively preserved. Additionally, asymmetry of muscle involvement in the lower legs was noted, as well as pronounced asymmetry in the clinical manifestation in the form of more severe muscle atrophy of the left hand and lower leg muscles. Previously, a similar case with an asymmetric atrophy of the left hand and lower arm muscles was reported in the literature, but the leg muscles were most preserved [17]. Another atypical case with predominant involvement of calf muscles was described later [16]. These two atypical cases and our patient No. 18.1 had different genotypes. At the same time, patients with the same *GNE* variants as in our patient No. 18.1 had no peculiar phenotypic features [12,31].

In conclusion, our study is the largest cohort of Russian patients with GNEM so far. Ten novel variants are presented expanding the known mutational spectrum of this myopathy. Our results showed that the novel missense variant c.169_170delGCinsTT (p.(Ala57Phe)) is the second most frequent variant in the patients from Russia. The most frequent variant in our cohort is previously reported in Roma and Rajasthan patients c.1853T>C (p.(Ile618Thr)). Most of our patients showed typical clinical picture of GNEM, although atypical features were noted in two patients.

## Figures and Tables

**Figure 1 genes-13-01991-f001:**
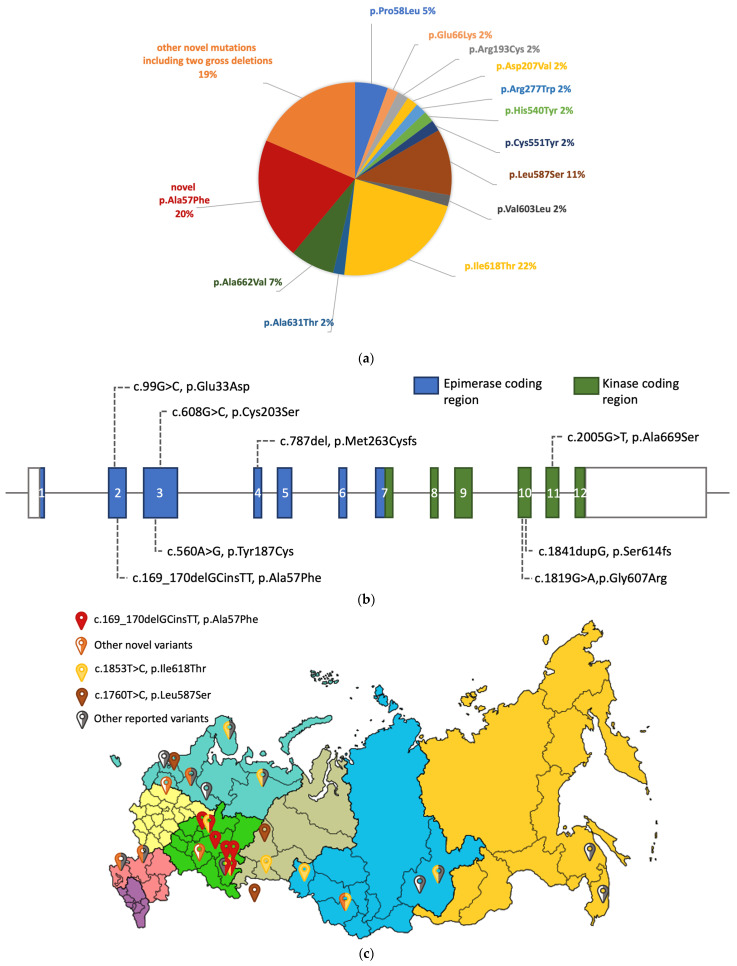
Mutational spectrum of GNEM in Russia. (**a**) Distribution of all detected variants; (**b**) Distribution of novel causative variants on the scheme of the *GNE* gene, and (**c**) Place of birth of 26 out of 27 unrelated patients with variants in the *GNE* gene are presented on the map of the Russian Federation. One patient was from Kazakhstan, but she identified her ancestry as Russian. Three variants were detected in a homozygous state (those families are identified as full-colored signs). Other variants were revealed only in a compound heterozygous state (those families are identified as half-colored signs). Only one variant, p.Ala57Phe, is clustered in neighboring regions of Russia.

**Figure 2 genes-13-01991-f002:**
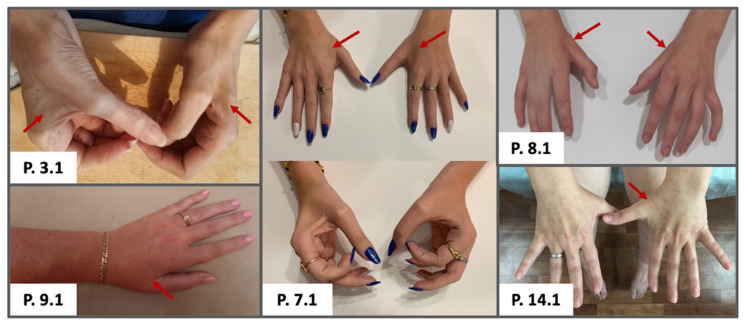
Hands of GNEM patients. Prominent atrophy of hand muscles, especially first dorsal interosseus, are noted with arrows.

**Figure 3 genes-13-01991-f003:**
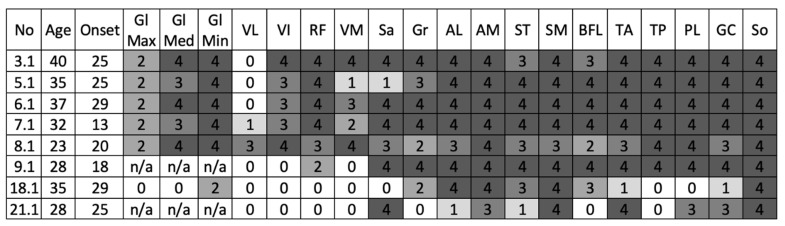
Muscle MRI data in the cohort of GNEM patients. The scores and colors represent the severity of fat infiltration of skeletal muscle according to the scale published by Mercuri et al. [23] Gl Max—gluteus maximus; Gl Med—gluteus medius; Gl Min—gluteus minimus; VL—vastus lateralis; VI—vastus intermedius; RF—rectus femoris; VM—vastus medialis; Sa—sartorius; Gr—gracilis; AL—adductor longus; AM—adductor magnus; ST—semitendinosus; SM—semimembranosus; BFL—biceps femoris caput longum; TA—tibialis anterior; TP—tibialis posterior; PL—peroneus longus; GC—gastrocnemius; So—soleus; n/a— not applicable.

**Figure 4 genes-13-01991-f004:**
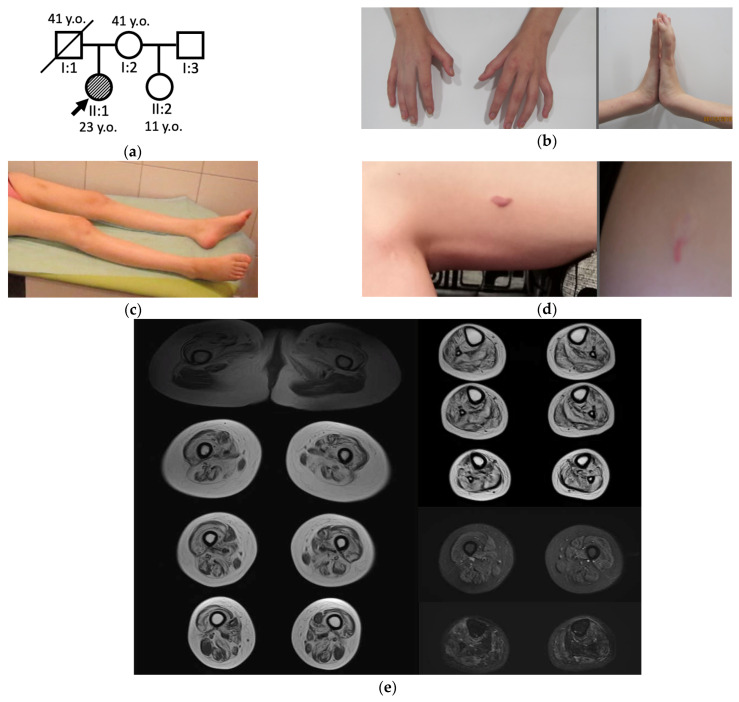
The patient No. 8.1. with atypical GNEM. (**a**) Genealogy; (**b**) Hand muscle atrophy and hand joint contractures; (**c**) Lower limb muscle atrophy; (**d**) Keloid scars on the left thigh and left arm; (**e**) T1-weighted and T2-STIR muscle MR-images of the lower limbs.

**Figure 5 genes-13-01991-f005:**
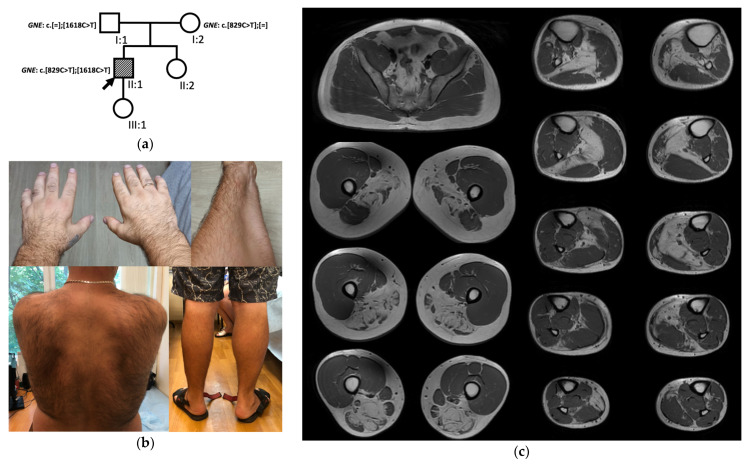
The patient No. 18.1. with atypical GNEM. (**a**) Genealogy; (**b**) Hands and left lower arm muscles’ atrophy, asymmetrical atrophy of lower leg muscles; (**c**) T1-weighted muscle MR-images of the pelvic girdle and lower limbs.

**Table 1 genes-13-01991-t001:** Novel variants in patients with GNEM from Russia.

cDNA (NM_001128227.3)	Protein	Type	Domain	Frequency in GnomAD	ACMG Classification
c.99G>C	p.(Glu33Asp)	Missense	Epimerase	absent	LPV (PM2, PM3, PM5, PP3)
c.169_170delGCinsTT	p.(Ala57Phe)	Missense	Epimerase	absent	LPV (PM2, PM3, PM5, PP3)
c.560A>G	p.(Tyr187Cys)	Missense	Epimerase	absent	VUS (PM2, PM3, PP3)
c.608G>C	p.(Cys203Ser)	Missense	Epimerase	0.000004	VUS (PM2, PM3, PP3)
c.1819G>A	p.(Gly607Arg)	Missense	Kinase	absent	VUS (PM2, PM5, PP3)
c.2005G>T	p.(Ala669Ser)	Missense	Kinase	absent	VUS (PM2, PM5, PP3)
c.1841dupG	p.(Ser614ArgfsTer6)	Frameshift	–	absent	PV (PVS1, PM2, PM3, PP3)
c.787del	p.(Met263CysfsTer4)	Frameshift	–	absent	PV (PVS1, PM2, PP3)

PV—pathogenic variant, LPV—likely pathogenic variant, VUS—variant of unknown significance.

**Table 2 genes-13-01991-t002:** Clinical features and molecular data of Russian patients with GNE myopathy.

No	Genotype (NP_001121699.1)	Age (y)/Gender	Age of Onset (y)	First Symptoms, Weakness	Amb/Age at Loss	GNEM-FAS (score)	Muscle Atrophy	30 MMT (MRC Score)	Neck Muscle Weakness	Atypical Features	CK (U/L)	Platelets (×10^9^/L)
**1.1**	p.(Glu66Lys); p.(Glu33Asp)	31/f	25	FD, low back pain	y	76	FA, H, LL, F	115	n	n	372	n/a
**2.1**	p.(Pro58Leu); p.(Ala631Thr)	33/f	28	FD	n/32	35	S, FA, H, LL, F	n/a	n	n	242	232
**3.1**	p.(Cys203Ser); p.(Ala662Val)	40/f	25	FD	n/36	4	S, FA, H, LL, F	n/a	y	n	614	n/a
**4.1**	p.(Val603Leu); p.(Ala662Val)	31/f	24	difficulties in climbing stairs	y	68	FA, H, LL, F	116	n	n	344	246
**4.2**	p.(Val603Leu); p.(Ala662Val)	32/m	29	difficulties in climbing stairs	y	85	FA, LL, F	n/a	n	n	n/a	n/a
**5.1**	p.(Ile618Thr); del ex1-ex3	35/m	25	FD	y	77	FA, H, LL, F	101	n	n	1050	213
**6.1**	p.(Ile618Thr); p.(Tyr187Cys)	37/f	29	FD	y	75	LL, F	114	y	n	629	292
**7.1**	p.(Ser614Argfs); p.(Ala662Val)	32/f	13	left leg muscles wasting	n/26	17	FA, H, LL, F	64	y	n	200	240
**8.1**	p.(Ile618Thr); del ex5-ex7	23/f	20	FD	y	n/a	S, FA, H, T, LL, F	90	y	joint contractures, kelloid scars	103	197
**9.1**	p.(Arg193Cys); p.(Ala662Val)	28/f	18	FD	n/26	27	FA, H, LL, F	70	y	n	250	222
**10.1**	p.(Met263Cysfs); p.(Ala669Ser)	40/m	23	left FD	n/37	24	S, FA, H, T, LL, F	n/a	y	n	1200	n/a
**11.1**	p.(Ala57Phe); p.(Ile618Thr)	35/f	27	FD	y	86	FA, H, LL, F	116	n	n	164	242
**12.1**	homozygous p.(Ala57Phe)	41/m	26	FD	y	48	S, FA, H, T, LL, F	70	y	n	n/a	n/a
**12.2**	homozygous p.(Ala57Phe)	38/m	23	FD	n/33	17	S, FA, H, T, LL, F	52	y	n	n/a	n/a
**12.3**	homozygous p.(Ala57Phe)	30/f	21	inability to stand on toes	y	73	FA, H, LL, F	119	n	n	n/a	n/a
**13.1**	homozygous p.(Ala57Phe)	41/f	20	arm muscles weakness, FD	n/30	2	S, FA, H, T, LL, F	14	y	n	49	n/a
**14.1**	homozygous p.(Ile618Thr)	40/f	21	FD	y	52	S, FA, LL, F	84	n	n	168	n/a
**15.1**	homozygous p.(Ile618Thr)	21/m	18	pain in leg muscles	y	93	LL, F	127	n	n	850	411
**16.1**	homozygous p.(Ile618Thr)	29/f	24	low back pain	y	n/a	H, LL, F	116	n	n	179	n/a
**17.1**	homozygous p.(Ala57Phe)	24/f	17	FD	y	n/a	LL, F	n/a	n	n	650	n/a
**18.1**	p.(Arg277Trp); p.(His540Tyr)	35/m	29	pain in leg muscles	y	91	H, LL, F	134	n	pronounced asymmetry	950	202
**19.1**	p.(Ile618Thr); p.(Pro58Leu)	29/m	25	FD	y	97	LL, F	138	n	n	1020	n/a
**19.2**	p.(Ile618Thr); p.(Pro58Leu)	29/m	21	FD	y	68	FA, H, LL, F	98	n	n	371	183
**20.1**	p.(Ala57Phe); p.(Cys551Tyr)	25/f	17	FD	y	38	H, LL, F	93	n	n	183	363
**21.1**	p.(Gly607Arg); p.(Gys203Ser)	28/f	25	difficulties in climbing stairs	y	81	H, LL, F	119	y	n	400	126

Amb—ambulatory; GNEM-FAS—GNE Myopathy Functional Activity Scale; 30 MMT—manual muscle testing of 30 muscles by using the Medical Research Council’s (MRC) score; CK—creatine kinase; FD—foot drop; S—shoulder; FA—forearm; H—hand; T—thigh; LL—lower leg; F—foot.

## Data Availability

The data that support the findings of this study are available from the corresponding author, [A.M.], upon reasonable request.

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
