# Peer review of "Genetic and Clinical Spectrum of GNE Myopathy in Russia"

_genes, 2022, doi:10.3390/genes13111991_

Round 1
Reviewer 1 Report
Thank you for inviting me to review this manuscript. The manuscript is well written and provides additional information to the spectrum of GNE myopathy phenotype and genotype, which would be of interest to neuromuscular clinicians with a special interest in distal myopathies.
I suggest considering the following revisions and clarifications:
1. Please state the status of the ethics committee revision and patient informed consent for this research, and also if applicable in the local settings, whether the data collection and storage follows GDPR guidance.
2. Please state the type of research (e.g. prospective, retrospective, cross-sectional, type of cohort) and whether there is any potential explanation of female predominance in this cohort.
3. 74-75 “Three of the unrelated patients were 75 Bashkirs, the rest of the patients were Russians.” Could you clarify what does it mean (e.g. ethnic origin or place of residence).
4. 77-78 “Neurological examination 77 was performed in 25 patients from 21 unrelated families.” I suppose this means that 6/31 (or 20% of the study population) were not examined therefore it leaves a gap in phenotyping and probably requires consideration of other causes of their symptoms.
5. 93 Could you clarify please how the patients were referred to the lab for genetic analysis? E.g. does the lab provides a national service or has a catchment area limited to a number of regions or is it a commercial setting with patients self-referring themselves?
6. 182-183 “Ankle 182 reflexes were reduced uniformly..” please clarify if this suggests a neurological deficit or muscle weakness in this case.
7. 194 “one patient had thrombocytopenia” – please provide the actual number with the refence range of normal limits in this case for comparison with the previously published cases.
8. 196 and 246 Do I understand correctly that SMA and hereditary neuropathy was a clinical diagnosis here and was based on a relevant phenotype only and was not genetically confirmed? Or is it a “double trouble case” of two conditions?
9. 325 “.. a more severe course of the disease in women”- would it be possible to clarify whether the muscle weakness decline was related to the pregnancies in these women?
10. There is no mentioning of other significant co-morbidities which can contribute to severity of the disease.
11. I assume there was no cardiac or respiratory compromise in these patients, but could you confirm that please
12. Finally, did any of the patients, especially ones with VUS, have muscle biopsy to confirm compatible presentation with GNE myopathy?
Author Response
Dear reviewer,
We highly appreciate your constructive suggestions and comments. They were very helpful in improving our manuscript. Our responses to the comments are given point-by-point below. All modifications in the manuscript have been highlighted in yellow.
- Study was approved by the local ethics committee at our center. In our practice we don't use GDPR guidance, and we manage personal information according to local laws.. All patients signed an informed consent, that included information about publication. For 25 patients with clinical data we obtained an additional informed consent at the time of examination.
- Our study is a retrospective cohort study of patients with genetically confirmed GNEM. We added that information in the paper (72). Because of the small size of the patient group, we cannot explain for sure why female patients are predominant. As mentioned in the discussion the disease course is usually more severe in females, probably that is the reason why female patients refer to DNA testing more frequently. However, we would not like to speculate about that in the paper due to the size of the group, because it also can be coincidence or related to different factors.
- We added the specification: “Three of the unrelated patients were from Bashkir ethnic group, the rest of the patients were of Russian origin” (76-77).
- Unfortunately, we couldn’t examine in-person 6 patients and didn’t collect enough clinical data about them for publication (we added that information in our text, 80-81). Initial clinical data that we received before exome sequencing is insufficient and low quality for discussion in this paper. All of them were referred to the genetic testing because of neuromuscular disorder, and all of them have earlier reported variants.
- Our center is a national healthcare provider for inherited diseases, so patients from all the regions could be referred for genetic counseling or send their biomaterials for various molecular testing by a local doctor.
- Absent ankle reflexes is a typical sign for GNEM patients due to calf muscle atrophy and weakness.
- We added the count of the platelets in the text (198-199): “One patient (No. 21.1) had thrombocytopenia with a platelet count of 126×109/L (nl 180-450×109/L)”.
- The patient No. 18.1. with atypical GNEM has been clinically diagnosed with SMA before getting relevant EMG and muscle MRI data because of a very unusual clinical picture. Before NGS study, the patient was referred to SMN1 copy analysis. He has two copies of that gene. He has quite moderate disease severity compared to other GNEM patients and we didn't find in NGS data any other clinically significant variants except for the GNE gene. That’s why we estimate the possibility of a double-trouble condition as unlikely . The corrections are inserted to the text (251-252 and 281-282).
- It is undoubtedly that a pregnancy is a very important factor exacerbating GNEM in women. Three of five female non-ambulant patients mentioned that the disease got worse after the delivery, but they did not associate their pregnancies with the ambulation loss.
- Only two patients mentioned co-morbid conditions as nodular goiter and hypothyroidism. Both patients' conditions were corrected. Other patients did not note any other chronic conditions.
- Yes, we didn’t observe any cardiac dysfunction in our patient group. One patient with a severe course of the disease has respiratory dysfunction. She is reported in our cohort. She complained of the difficulty breathing in the supine position on a low pillow. She had no opportunity to perform respiratory tests due to her ambulant status.
- Absence of muscle biopsy data is one of the biggest limitations of our paper. Unfortunately, no one of our patients had a proper morphological study of a muscle biopsy due to the low availability of muscle biopsy in Russia.
Reviewer 2 Report
This study is highly descriptive of GNE Myopathy genetic and clinical picture in patients from Russia. The paper is well written. There is little real new clinical information apart from the first 'unusual case', so this should be treated with caution as there are too many atypical features. Is there another single heterozygous mutation in a collagen related muscle disease? Please report.
What was the value of the thrombocyte count in the single patient with thrombocytopenia and what mutation did he/she have- was there a bleeding tendency or just a lab abnormality? Ten novel mutations are described in this cohort, making it highly heterogeneous.
This is a very informative picture of GNE Myopathy in Russia .
Author Response
Dear reviewer,
We are very grateful for your precious time in reviewing our paper and providing valuable comments. Below we provide the point-by-point responses. All modifications in the manuscript have been highlighted in yellow.
- Yes, at first we checked variants in the COL6 genes. We did not find any appropriate SNV or CNV by coverage analysis.
- We added that information in the text: “One patient (No. 21.1) had thrombocytopenia with a platelet count of 126×109/L (nl 180-450×109/L)”. The platelet count was mildly decreased and was not accompanied by any clinical abnormalities.